# Poisson’s Ratio Prediction of Injection Molded Thermoplastics Using Differential Scanning Calorimetry

**DOI:** 10.3390/polym16141956

**Published:** 2024-07-09

**Authors:** Tetsuo Takayama, Yuuki Nagasawa

**Affiliations:** 1Graduate School of Organic Materials Science, Yamagata University, Yonezawa 992-8510, Japan; 2Faculty of Engineering, Department of Polymeric and Organic Materials Engineering, Yamagata University, Yonezawa 992-8510, Japan

**Keywords:** injection molding, mechanical properties, free volume

## Abstract

In the development of thermoplastic products, it is necessary to conduct the necessary mechanical tests and evaluate the reliability of thermoplastics in each case, because the mechanical properties of the same material vary depending on the molding process conditions and product shape. In order to build a sustainable society, it is expected that the evaluation of the mechanical properties of thermoplastics, which are resource and energy saving, will be required. In this paper, the glass transition temperature and melting point of injection-molded thermoplastics were evaluated by differential scanning calorimetry, and the correlation between Poisson’s ratio and free volume was obtained by applying the theory proposed by Flory et al. A certain correlation was found between the Poisson’s ratio of polymers and the change in free volume determined by the glass transition temperature. It is also clear that this relationship can be approximated by orders of magnitude. The Poisson’s ratio of the core layer tended to be smaller than that of the skin layer. It has also been found that there is a negative correlation between the Young’s modulus and the free volume of the polymer material.

## 1. Introduction

Thermoplastics are employed in a multitude of fields, from the mundane to the sophisticated, due to their lightweight nature and superior moldability when compared to metals and ceramics [1,2]. Despite the perception that thermoplastics have low energy costs and a minimal environmental impact due to their low melting temperatures, their waste is regarded as an environmentally polluting material [3,4,5]. One reason for this is that thermoplastics deteriorate as they are used as products, making it difficult to recycle their waste as a resource. Additionally, the mechanical properties of thermoplastic products can vary significantly depending on the molding and processing conditions and the shape of the product. Therefore, it is necessary to conduct the necessary mechanical tests each time to evaluate their reliability [6,7]. The fabrication of dedicated molds for mechanical testing entails three costs: materials, time, and energy. In the case of metals and ceramics, mechanical properties are largely unaffected by molding process conditions and product shape. Consequently, the cost of developing a thermoplastic product is not necessarily lower than that of metals and ceramics. In light of the aforementioned circumstances, it is anticipated that the assessment of the mechanical properties of thermoplastics, which are both resource- and energy-efficient, will be a crucial aspect in the formation of a sustainable society.

One of the mechanical properties is the modulus of elasticity. The modulus of elasticity is a general term used to describe the relationship between strain and stress that occurs when an external force is applied to an elastic body. In particular, Young’s modulus, shear modulus, bulk modulus, and Poisson’s ratio are applicable.

This study focuses on Poisson’s ratio. Poisson’s ratio is the ratio of longitudinal strain ε_l_ to transverse strain ε_t_ caused by uniaxial tensile loading and is described by the following Equation (1).
(1)$$\upsilon =-\frac{\varepsilon_{t}}{\varepsilon_{l}}$$

In the above equation, a minus sign is added to indicate that Poisson’s ratio is positive when elongational deformation occurs and shrinkage occurs in its orthogonal direction. The bulge method [8] for films, the moiré interferometry [9] for bulk materials, the strain gauge method [10], and the ultrasonic method [11] are known methods for evaluating Poisson’s ratio. While these evaluation methods are effective for metals and ceramics, their applicability is limited for polymeric materials. For instance, in the strain gauge method, it is challenging to accurately assess strain when the material is soft, such as an elastomer, due to the influence of the strain gauge’s stiffness on the strain measurement. Similarly, in the ultrasonic method, evaluating the velocity of the input wave can be challenging due to the significant vibration damping caused by viscoelasticity.

The authors proposed a method for evaluating Poisson’s ratio using a three-point bending test [7]. This method combines the fact that the deformation induced by three-point bending loading is pure elongation deformation with the authors’ proposed theory of the stress at shear yield initiation of polymers to obtain the elastic modulus of the skin layer region. Furthermore, the authors reported that the application of a short-beam shear test can apply triaxial shear stress to the core layer region [12]. They also found that the composite component of the yield-initiating shear stress in each direction obtained by this method is the yield-initiating shear stress when the molded product is subjected to uniaxial tensile loading. This implies that the orientation of the shear plane of the molded product can be determined using this method. By employing this relationship, it is anticipated that the Poisson’s ratio of each method can be obtained. In other words, the Poisson’s ratio of the skin layer region can be determined by the three-point bending test and the Poisson’s ratio of the core layer region can be determined by the short-beam shear test. This is beneficial for comprehending the mechanical properties of each layer of an injection-molded product that is intended to form a hierarchical structure.

Figure 1 depicts a model of Poisson deformation. In the absence of cross-linking, the model in Figure 1 can be utilized to elucidate Poisson deformation. In this scenario, the potential range of Poisson’s ratios is delineated by the following Equation (2).
(2)0<υ<0.5

In both models, a space exists in the center of the model that allows for Poisson deformation. In this study, this space is considered to be the free volume. The concept of free volume is explained, for example, by the lattice point model of Figure 2 [13,14]. In this model, one polymer chain is represented by a series of segments (●), and it is assumed that these segments can occupy one lattice point. An empty lattice point may arise when a polymer segment fills an empty lattice point without a gap. These empty lattice points are referred to as free volume. It is postulated that the existence of sufficient free volume allows molecular diffusion, and, thus, free volume is thought to be involved in the magnitude of Poisson deformation.

It has been demonstrated that the free volume of a substance correlates with its glass transition temperature. For instance, Flory demonstrated that the following Equation (3) is valid between the glass transition temperature and the free volume content [13,14].
(3)$$f=\alpha \left(T-T_{g}\right)+f_{0}=\Delta f+f_{0}$$

Here, f_0_ indicates the free volume content below the glass transition temperature T_g_, which is said to be constant at 0.025. Additionally, α is the coefficient of thermal expansion. Given that the difference between the target temperature T and the glass transition temperature is correlated with the free volume from Equation (3), it is anticipated that a quantitative correlation between the free volume and Poisson’s ratio can be obtained if the glass transition temperature is obtained and T is appropriately defined. In injection molding, a polymer melt is injected into a mold, formed, and then cooled and solidified to form a molded product. The cooling process occurs rapidly in the skin layer region, which is the first area in contact with the mold. Therefore, the free volume in the skin layer region is considered to be equivalent to that of the melt. This implies that T in Equation (3) can be regarded as the melt processing temperature. In other words, it is anticipated that the correlation between Poisson’s ratio and free volume can be quantitatively determined in the skin layer region of injection molded products.

In this study, the glass transition temperature of injection-molded thermoplastics was determined by differential scanning calorimetry. The free volume was determined using Flory’s theory, and the Poisson’s ratio of the skin layer was determined from the results of a three-point bend test to investigate the correlation between the two. Furthermore, the Poisson’s ratio of the core layer was derived from the outcomes of short-beam shear tests. Additionally, the distribution of free volume in thermoplastic injection-molded products was evaluated by comparing the results of the Poisson’s ratio of the skin layer and core layer.

## 2. Materials and Methods

### 2.1. Materials

This study employed a range of amorphous polymers, including one type of polystyrene (PS), three types of polycarbonate (PC) with varying viscosities, and one type of acrylonitrile-butadiene-styrene copolymer (ABS). One homo-type polypropylene (Homo-PP), two highly rigid types of polypropylenes (HM-PP) with different catalysts, one block-type polypropylene (Block-PP), and one polyoxymethylene (POM) were employed as crystalline polymers. Details of each material are presented in Table 1. The table also shows the melt volume flow rate (MVR), which is an index of melt viscosity dependent on molecular weight.

### 2.2. Sample Preparation

The polymer pellets were filled into an ultra-compact electric injection molding machine (C, Mobile0813; Shinko Sellbic Co., Ltd., Tokyo, Japan), and were injection molded according to the conditions shown in Table 2. The table lists the following parameters: T_inj_ (injection molding temperature), T_mold_ (mold temperature), V_inj_ (injection speed), Ph (holding pressure), t_inj_ (filling holding time), and t_cool_ (cooling time). In this study, the injection speed was fixed at 30 mm/min, and three different molding process temperatures were investigated. Figure 3 depicts the shape of the molded product. The molded product exhibited a beam-shaped morphology with a width of 5 mm, a thickness of 2 mm, and a length of 50 mm [15].

### 2.3. Glass Transition Temperature Determination

The surface area of the molded product obtained by injection molding was excised using a microtome (RX-860, Yamato Kohki Industrial Co., Ltd., Saitama, Japan), and differential scanning calorimetry (DSC, Q-2000; TA instruments Co., Ltd., New Castle, DE, USA) was performed on the obtained samples. Measurements were taken in the temperature range from −40 to 230 degrees Celsius, and the heat-flow curve during the temperature increase process was examined. The temperature increase rate was fixed at 10 K/min. Measurements were taken in an atmosphere displaced by nitrogen at a flow rate of 50 mL/min.

The results obtained from the amorphous polymer moldings were analyzed according to ISO 11357-2:2020, and the region where the base shift occurred was identified. The intermediate glass transition temperature was then determined from this region as the glass transition temperature [16]. The glass transition temperature, T_g_, was determined using the differential scanning calorimetry (DSC) method. The free volume was then calculated using Equation (3), with α set to 0.00048 [1/K], as previously reported in the literature.

Conversely, the results of the crystalline polymer moldings indicated that the melting point T_m_ was evaluated as the temperature of the peak value of the endothermic reaction that occurs during the melting of the crystals, in accordance with ISO 11357-3:2018 [17]. Furthermore, T_g_ was obtained according to the following Equation (4) [18,19].
(4)$$T_{g}=BT_{m}$$

The coefficient B was applied to PP and POM, which are asymmetric vinyl polymers, with a value of ⅔ [18,19]. Furthermore, the endothermic enthalpy of melting, ΔH_m_, was determined, and the crystallinity, X_c_, was obtained according to the following Equation (5).
(5)$$X_{c}=\frac{∆H_{m}}{∆H_{m0}}$$

The endothermic enthalpy, ΔH_m0_, is the quantity obtained at 100% crystallinity, with values of 207 J/g for PP and 326 J/g for POM [20]. The free volume of the crystalline polymer, f_c_, was determined using the following Equation (6), based on the values of T_g_ and X_c_ obtained from Equations (4) and (5).
(6)$$f_{c}=\alpha \left(T-T_{g}\right)\left(1-X_{c}\right)+f_{0}$$

The preceding equation modifies Equation (3) by assuming that the free volume of the crystalline region remains constant as the temperature increases, while the free volume of the amorphous region undergoes a change. T is the injection molding temperature, T_inj_, and the free volume changes very little before solidification is complete because the molten resin is frozen rapidly the moment it touches the mold.

### 2.4. Poisson’s Ratio at Skin Region

A three-point bending test was conducted on the prepared injection-molded product. The test was performed using a universal mechanical testing machine (MCT-2150; A&D Co., Ltd., Tokyo, Japan) with a loading speed of 10 mm/min. The span was set at 40 mm, with the molded product placed flatwise. The obtained load-deflection curves were utilized to generate bending stress, σ, –bending strain, ε_f_, curves in accordance with the following Equations (7) and (8), as outlined in ISO 178 [21].
(7)$$\sigma =\frac{3PS}{2wh^{2}}$$
(8)$$\varepsilon_{f}=\frac{6\delta h}{S^{2}}$$

In the aforementioned equation, the symbols P, δ, S, h, and w represent the load, deflection, span, specimen thickness, and specimen width, respectively. The maximum bending stress observed in the obtained bending stress–bending strain curve was identified as the flexural strength σ_f_, while the slope at the initial loading was determined to be the flexural modulus E_f_.

The deformation up to the onset of yielding in the tensile region due to three-point bending loading when installed flatwise can be approximated as pure extensional flow deformation. This approximation is made on the assumption that the molded product yields in its entire cross-section due to three-point bending loading. The relationship between σ_f_ and the yield initiation stress σ_y_ is expressed by the following Equation (9).
(9)$$\sigma_{f}=\frac{3}{2}\sigma_{fy}=\frac{3}{2}\sigma_{y}\left(1+\upsilon \right)$$
where υ is Poisson’s ratio in the elongation direction. The relationship between E_f_ and the Young’s modulus E is expressed by the following Equation (10) [7].
(10)$$E_{f}={\left\{\frac{\left(1+\upsilon \right)\left(1-2\upsilon \right)}{1-\upsilon }+\frac{3}{2}{\left(\frac{h}{S}\right)}^{2}\frac{\left(1+\upsilon \right)\left(1-2\upsilon \right)}{1-\upsilon }\right\}}^{-1}E$$

On the other hand, the authors believe that the onset of polymer yielding originates from molecular friction, and σ_y_ is expressed by the following Equation (11).
(11)$$\sigma_{y}=C\beta \Delta TE\cos\theta$$

In the above equation, β represents the average linear expansion coefficient between the forming process temperature and the test temperature. ΔT is the difference between the forming process temperature and the test temperature. C is a coefficient, with a value of $\sqrt{3}$ for the Mises yield condition and 2 for the Tresca condition [22]. θ is the shear angle, which is expressed by the following Equation (12).
(12)$$\theta =\tan^{-1}2\upsilon$$

Substituting Equations (11) and (12) into Equation (10) results in the values of variables other than ν becoming fixed. Consequently, when the items related to ν are assembled on the left-hand side and the others on the right-hand side, Equation (13) is formed.
(13)$$\frac{{2\sigma }_{f}}{3C\left(1+\upsilon \right)\beta \Delta TE_{f}}=\left\{\frac{\left(1+\upsilon \right)\left(1-2\upsilon \right)}{1-\upsilon }+\frac{3}{2}{\left(\frac{h}{S}\right)}^{2}\frac{\left(1+\upsilon \right)\left(1-2\upsilon \right)}{1-\upsilon }\right\}\cos\theta$$

In this study, the Poisson’s ratio υ_skin_ of the skin layer was estimated using the following equation, based on the results obtained from three-point bending tests. Five replicates of the three-point bending tests were conducted for each sample, and the mean value was employed as the representative physical property value.

### 2.5. Poisson’s Rario at Core Region

A short-beam shear test was conducted on the injection-molded parts prepared for the experiment. The short-beam shear test is a three-point bending test in which the span is intentionally narrowed [12]. Previous studies have reported that the stresses applied in this test are iso-triaxial shear stresses. The test was conducted using a universal mechanical testing machine (MCT-2150; A&D Co., Ltd., Tokyo, Japan) with a loading speed of 10 mm/min and a span of 10 mm, with the molded product placed edgewise. The average shear stress was obtained from the loads obtained according to the following Equation (14).
(14)$$\tau =\frac{3P}{4wh}$$

The obtained load-deflection was differentiated by deflection to obtain the stiffness, and a stiffness-averaged shear stress curve was generated. Figure 3 illustrates the aforementioned curve. It is evident that there are three points in the middle of the curve where the stiffness undergoes a discontinuous change, as indicated by the arrows. It can be postulated that shear yielding has occurred within the molded product at these three points. Consequently, the average shear stresses at these points are defined as τ_s_, τ_m_, and τ_l_, respectively, in decreasing order of value. The authors have reported a relationship between these shear stresses and the shear stress at the onset of yielding, τ_y_ [12].
(15)$$\tau_{y}=\sqrt{\tau_{s}^{2}+\tau_{m}^{2}+\tau_{l}^{2}}$$

This implies that the composite vector of τ_s_, τ_m_, and τ_l_ is oriented in the same direction as the shear plane. By defining the shear angles of the planes as the orientations of the projection of the shear planes onto the respective planes, the following shear angles can be obtained.
(16)$$\theta =\tan^{-1}\frac{\tau_{s}+\tau_{m}}{2\tau_{l}}$$

Here, for injection molded products, the flow direction (MD) is τ_l_, the width direction (TD) is τ_m_, and the thickness direction (ND) is τ_s_ [12]. To summarize the above relationships, the Poisson’s ratio υ_core_ of the core layer can be obtained using τ_s_, τ_m_, and τ_l_ by the Equation (17).
(17)$$\upsilon_{\mathrm{core}}=\frac{\tau_{s}+\tau_{m}}{4\tau_{l}}$$

This paper compares the free volume present in the skin and core layers by determining the correlation between the υ_core_ obtained in Equation (17) and the υ_skin_ obtained from the results of a three-point bending test. Five replicates of the short-beam shear tests were conducted for each sample, and the mean value was employed as the representative physical property value.

## 3. Results

### 3.1. Relationship between Poisson’s Ratio and Free Volume of Amorphous Polymers

Table 3 presents the flexural strength (F.S.) and flexural modulus (F.M.) of amorphous polymers obtained through the three-point bending test, along with the υ_skin_ estimated using the methodology described in Section 2.4. The same table also displays the glass transition temperature obtained through differential scanning calorimetry (DSC). The change in free volume, Δf, was obtained by a modification of Equation (3). The rationale for expressing it as Δf is to provide a precise indication of the specific items involved in the Poisson deformation. The model depicted in Figure 1 indicates that Poisson deformation is possible due to the existence of gaps between molecules. The f0 in Equation (3) is a constant that is independent of temperature. Although a range of values for this constant is possible, this range is considered to originate from the excluded volume effect and is therefore not involved in the Poisson deformation. No clear correlation was observed between the injection molding temperature and the glass transition temperature. This suggests that the change in free volume is mainly derived from the injection molding temperature.

Conversely, υ_skin_ tends to increase with increasing injection molding temperature. Figure 4 illustrates the relationship between υ_skin_ and the free volume of amorphous polymers, which is calculated from the glass transition temperature. Here, since the change in free volume was primarily derived from the injection molding temperature, as evidenced by the results in Table 3, The free volume f_0_ that exists below the glass transition temperature was excluded and used as Δf on the horizontal axis. This figure revealed a correlation between υ_skin_ and Δf for amorphous polymers. Furthermore, this correlation can be approximated by the following Equation (18).
(18)$$\upsilon =A\Delta f^{B}$$
where A and B are constants; for amorphous polymers, A was 1.0 and B was 0.38.

### 3.2. Relationship between Poisson’s Ratio and Free Volume of Crystalline Polymers

Table 4 presents the flexural strength and flexural modulus of the crystalline polymers obtained by the three-point bending test, along with the υ_skin_ estimated by the method described in Section 2.4. The same table also shows the melting point determined by DSC and the free volume determined by Equations (3) and (19).
(19)$$\Delta f_{c}=f_{c}-f_{0}=\alpha \left(T-T_{g}\right)\left(1-X_{c}\right)$$
where X_c_ is the crystallinity. The melting point of PP exhibited an upward trend with increasing injection molding temperature, while the crystallinity exhibited a downward trend. This phenomenon may be attributed to the relatively high melt viscosity at low injection molding temperatures, which results in strong molecular orientation formation in the skin layer region. No clear correlation was observed for POM. Despite the aforementioned changes, the upward trend in Δf with increasing injection molding temperature remained unchanged regardless of material. This suggests that the change in free volume is primarily influenced by the injection molding temperature. Conversely, the Poisson’s ratio tends to increase with elevated injection molding temperatures. Based on these findings, it can be postulated that there is a correlation between υ_skin_ and the free volume of crystalline polymers. Figure 5 illustrates a correlation between υ_skin_ and Δf for crystalline polymers, although the discrepancies between materials are more pronounced. This suggests that it is necessary to exclude the crystalline region when determining the free volume of crystalline polymers. Figure 6 shows the relationship between υ_skin_ of crystalline polymers and Δf_a_, which is the free volume f_a_ minus f_0_, obtained from Equation (19). This figure shows that the relationship between Poisson’s ratio and free volume for crystalline polymers can be represented by a single curve for the region excluding the crystalline region. A single curve can be generated for the region excluding the crystalline region, similar to the result for amorphous polymers. Furthermore, this correlation can be approximated by Equation (18), as was done for amorphous polymers, with A and B values of 1.2 and 0.40, respectively, and coefficients similar to those obtained for amorphous polymers.

### 3.3. Poisson’s Ratio of the Core Layer Region of Amorphous Polymer Injection Molded Products

Table 5 presents the values of τ_s_, τ_m_, and τ_l_ obtained from short-beam shear tests of amorphous polymers, along with the υ_core_ values calculated using Equation (17).

Figure 7 depicts the relationship between υ_core_ and υ_skin_, demonstrating that in the case of amorphous polymers, the Poisson’s ratio tends to be smaller in the core layer than in the skin layer.

### 3.4. Poisson’s Ratio of the Core Layer Region of Crystalline Polymer Injection Molded Products

Table 6 presents the values of τ_s_, τ_m_, and τ_l_ obtained from short-beam shear tests of amorphous polymers, along with the υ_core_ values derived from these data using Equation (17). Figure 8 depicts the relationship between υ_core_ and υ_skin_, demonstrating that in the case of crystalline polymers, the Poisson’s ratio tends to be smaller in the core layer than in the skin layer.

## 4. Discussion

### 4.1. Free Volume Distribution of Injection Molded Products

The relationship between the Poisson’s ratios of amorphous and crystalline polymers, as illustrated in Section 3, indicates that the Poisson’s ratio of the core layer is less than that of the skin layer in injection molded products. This implies that the free volume of the core layer is smaller than that of the skin layer when evaluated according to Equation (17). The rationale behind this phenomenon is elucidated in the context of the solidification behavior that occurs during injection molding. In injection molding, the molten polymer is filled into the mold to produce a molded product. At this stage, the molten polymer cools from the area in contact with the mold inwardly. This implies that the cooling and solidification completion time differs between the skin layer and the core layer. The skin layer and the core layer exhibit distinct characteristics. The longer the cooling and solidification completion time, the more readily the molecular orientation due to the pressure received during the mold filling and pressure retention stages can relax. Relaxation causes the molecules to move in the direction of a smaller free volume, resulting in a smaller free volume of the core layer compared to the skin layer.

The results presented in this paper can be interpreted as a Poisson’s ratio reproduction of the phenomenon described in this discussion. If we assume that the relaxation in the core layer proceeds by the mechanism described above and consider the results in Equations (3) or (19), we can reproduce the above discussion by interpreting that T in the equation has become smaller in the core layer. In other words, if the change in T due to relaxation is Tr, this value can be expressed by the following Equation (20).
(20)$$T_{r}=\frac{f_{skin}-f_{core}}{\alpha }=\frac{1}{\alpha }\left\{\sqrt[{B}]{\frac{\upsilon_{skin}}{A}}-\sqrt[{B}]{\frac{\upsilon_{core}}{A}}\right\}$$

This value can be obtained from the difference between the free volume of the skin layer, f_skin_, and the free volume of the core layer, f_core_, i.e., the difference between υ_skin_ and υ_core_. The right-hand side of Equation (20) is in Poisson’s ratio form using Equation (18). The results of T_r_ obtained from Equation (20) are shown in Table 7. From this table, it can be seen that T_r_ tends to increase as the injection molding temperature increases. This trend indicates that there is minimal change in the amount of free volume in the core layer as the injection molding temperature increases, the skin layer region is strongly affected, and the skin layer is the region that controls the surface mechanical properties. Therefore, the injection molding temperature is an important factor in controlling surface mechanical properties.

Conversely, if the difference between the injection molding temperature and the mold temperature is significant, the cooling rate is anticipated to be high. Consequently, mold temperature is also regarded as a crucial factor in regulating surface mechanical properties. Given that the scope of this study did not encompass the mold temperature dependence, we intend to address this topic in the future.

### 4.2. The Free Volume of Crystalline Polymers

This paper derives the relationship between υ_skin_ and Δf_c_ based on the assumption that the change in free volume of crystalline polymers with increasing temperature occurs only in the amorphous region. Furthermore, it is shown that this relationship is roughly consistent with the same relationship for amorphous polymers. The reason for this is discussed in terms of the difference in elastic moduli between the crystalline and amorphous regions. It is known that the elastic modulus of the crystalline region is two orders of magnitude higher than that of the amorphous region [23,24]. Furthermore, since the elastic modulus is negatively correlated with the coefficient of thermal expansion, the higher the elastic modulus, the less likely the material is to undergo thermal expansion. For example, Barker et al. experimentally demonstrated that there is a relationship between the elastic modulus and the linear expansion coefficient in the following Equation (21) [25].
(21)$$\alpha ≅\sqrt{\frac{15}{E}}$$

In Equation (21), E represents Young’s modulus. According to this equation, when the elastic modulus increases by two or more orders of magnitude, the linear expansion coefficient decreases by one or more orders of magnitude, and the increment of free volume in the crystalline region is one or more orders of magnitude smaller than that in the amorphous region. This implies that the increase in free volume due to heating is less likely to occur. When the elastic modulus is high, it can be concluded that, although the free volume of the crystalline region may also increase due to heating, the amount of increase is negligible and can be disregarded when calculating the free volume increment Δf. Based on the above discussion, it can be reasonably proposed that the relationship between Poisson’s ratio and free volume of crystalline polymers can be summarized in terms of Δf_c_, as proposed in this paper.

### 4.3. Relationship between Young’s Modulus and Free Volume of Polymers

The results of this paper demonstrate that free volume can be estimated by DSC measurement of injection-molded parts, and, furthermore, Poisson’s ratio can be predicted. Poisson’s ratio is one of the most important elastic moduli for determining the elastomechanical properties of solids. According to elastodynamics, for an isotropic solid, if two moduli can be identified, the other moduli can be determined. In other words, if Poisson’s ratio and one additional elastic modulus can be identified, all elastic moduli can be identified. Once the elastic moduli are identified, the yield initiation stress and fracture toughness can be determined using the mechanical model proposed by the authors [7]. However, the above discussion has not yet proposed another model that can identify the modulus of elasticity.

In addition to Poisson’s ratio, there are three elastic moduli: Young’s modulus, shear modulus, and bulk modulus. Figure 9 illustrates the relationship between Young’s modulus and Δf or Δf_c_. The Young’s modulus is obtained from Equation (10) as shown in Section 2.4. This figure shows that the Young’s modulus of polymeric materials tends to decrease as the free volume increases. However, the degree of decrease seems to vary from material to material, and it is difficult to predict the Young’s modulus from the free volume, as is the case with Poisson’s ratio, regardless of the material. This factor is discussed in the context of secondary bonding forces; The van der Waals’ force acts between the side chain directions of linear polymers to constrain the molecules to each other. There are other bonding forces, such as the hydrogen bond and the ionic bond, collectively referred to as the secondary bonding force. This bonding force is considerably smaller than that of the covalent bonding force. In other words, polymeric materials are those that are bonded by strong covalent bonds in the length direction of the molecular chain. Additionally, weak secondary bonding forces are present in the vertical direction of the molecular chain. The manner in which the molecules are packed is determined in order to minimize the free energy per unit volume, which is influenced by the conformation resulting from intramolecular rotation, the shape of the entire molecule, including side chains, and the type of intermolecular force [26]. During this process, crystalline polymers undergo the formation of lamellae with a folding chain structure.

The authors posit that the origin of the internal force generated when a polymeric solid is subjected to an external force is the frictional force generated between molecules. Based on this assertion, the intrinsic factor of the elastic modulus of polymeric materials can be considered to be the secondary bonding force that occurs in the vertical direction of the molecular chain or between molecules. The magnitude of this force determines the magnitude of the resistance force against deformation. Young’s modulus is the origin of the internal force generated when the polymer is subjected to elongation and deformation by an external force. Therefore, the secondary bonding force can be considered as the origin of the Young’s modulus of polymer solids. The magnitude of this depends on the intermolecular distance in addition to the conformation due to intramolecular rotation, the shape of the entire molecule including side chains, and the type of intermolecular force. Given that the intermolecular distance is positively correlated with the size of the free volume, it can be assumed that Young’s modulus tends to decrease as the free volume increases. Unlike Poisson’s ratio, the degree of decrease differs depending on the material. This is believed to be due to differences in conformation resulting from intramolecular rotation, the shape of the entire molecule including side chains, and the type of intermolecular force [26].

From the preceding discussion, it is determined that there is a negative correlation between free volume and Young’s modulus. However, the degree of dependence varies from molecule to molecule. Consequently, it is challenging to estimate two elastic moduli of polymeric materials by DSC measurement alone. In general, Young’s modulus can be obtained from uniaxial tensile tests. As demonstrated in this paper, Poisson’s ratio can be estimated using DSC measurement results. This enables the shear modulus and bulk modulus of the material to be evaluated by combining the Young’s modulus results obtained from uniaxial tensile tests for isotropic materials. This facilitates the intrinsic mechanical analysis of polymeric materials. Furthermore, if the Poisson’s ratio can be evaluated, it will be possible to estimate the free volume, which is important for understanding the structure of polymer solids. The results of this paper have significant implications for the precise development of the mechanics of polymers.

## 5. Conclusions

This paper evaluates the glass transition temperature and melting point of thermoplastic injection molded products by differential scanning calorimetry. It also applies the theory proposed by Flory et al. to obtain a correlation between Poisson’s ratio and free volume. The results are as follows.

A correlation was identified between the Poisson’s ratio of amorphous polymers and the free volume variation obtained using the glass transition temperature. Additionally, it was demonstrated that this relationship can be approximated by a power law.

A correlation was identified between the Poisson’s ratio of crystalline polymers and the increment in free volume determined using the glass transition temperature. By neglecting the increment in the crystalline region, a similar relationship was obtained for amorphous polymers.

The Poisson’s ratio of the core layer was observed to be smaller than that of the skin layer. This may be attributed to the fact that solidification occurs from the skin layer in contact with the mold, while the core layer region takes longer to solidify, and relaxation occurs during the solidification process.

A negative correlation was observed between the Young’s modulus and free volume of polymeric materials.

The findings of this study indicate that the Poisson deformation of thermoplastic polymers is primarily influenced by free volume, which is not subject to the excluded volume effect.

## Figures and Tables

**Figure 1 polymers-16-01956-f001:**
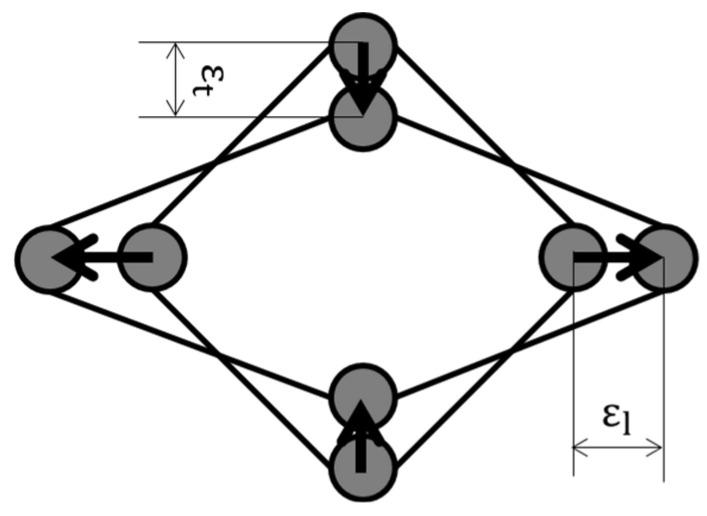
Poisson’s deformation model.

**Figure 2 polymers-16-01956-f002:**
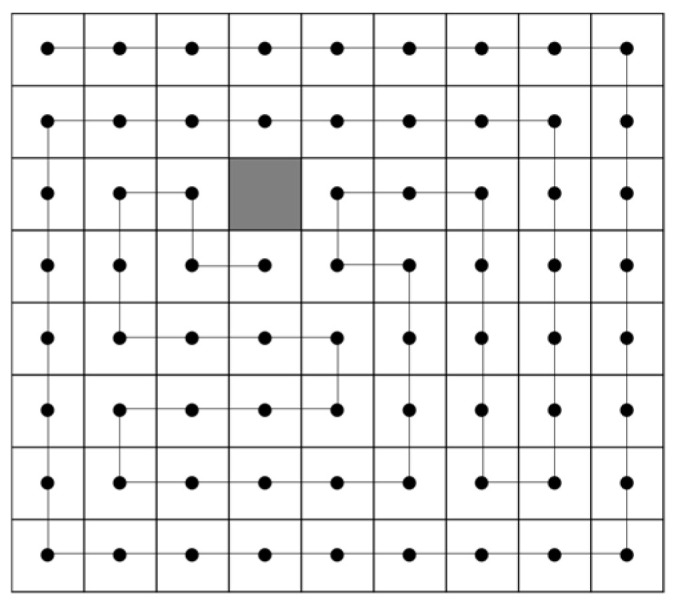
Free volume model. The gray area represents the free volume, while the black dot and line indicate the molecular chain.

**Figure 3 polymers-16-01956-f003:**
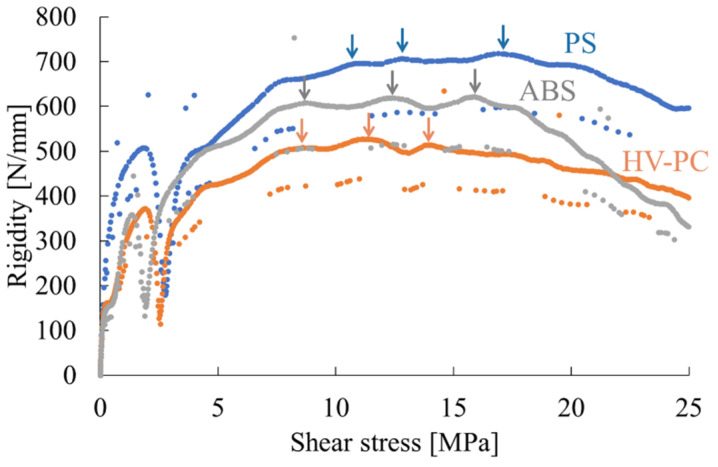
Examples of rigidity-averaged shear stress curves obtained from short-beam shear tests. Arrows in the figure means shear yield point at either direction.

**Figure 4 polymers-16-01956-f004:**
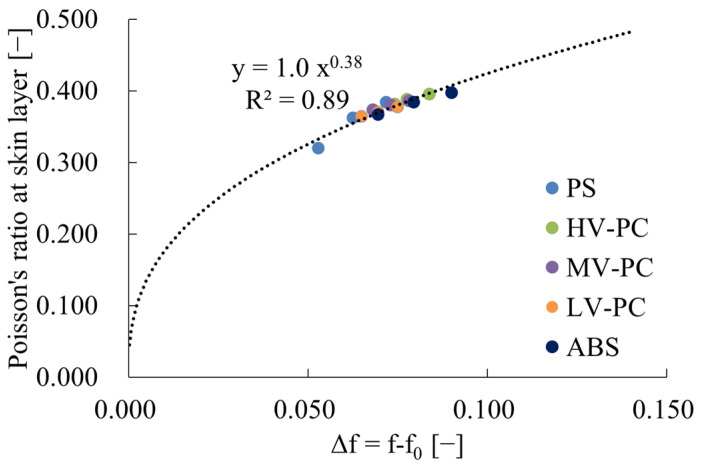
Relations between Poisson’s ratio at skin layer and Δf of amorphous polymer. R^2^ in the figure indicates the correlation coefficient.

**Figure 5 polymers-16-01956-f005:**
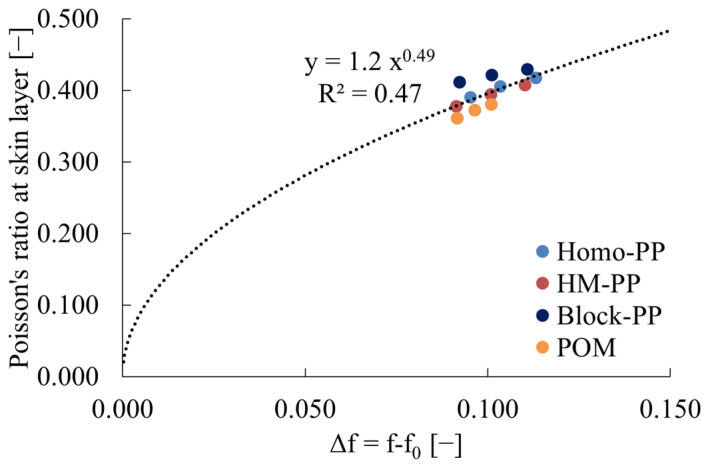
Relations between Poisson’s ratio at skin layer and Δf of crystalline polymers. R^2^ in the figure indicates the correlation coefficient.

**Figure 6 polymers-16-01956-f006:**
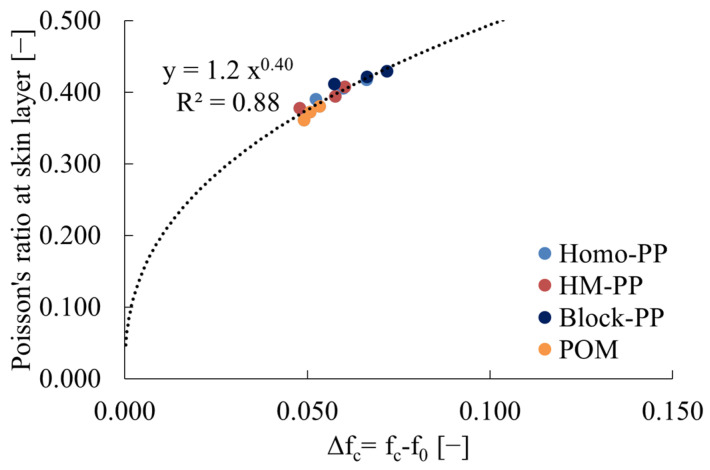
Relations between Poisson’s ratio at skin layer and Δf_c_ of crystalline polymers. R^2^ in the figure indicates the correlation coefficient.

**Figure 7 polymers-16-01956-f007:**
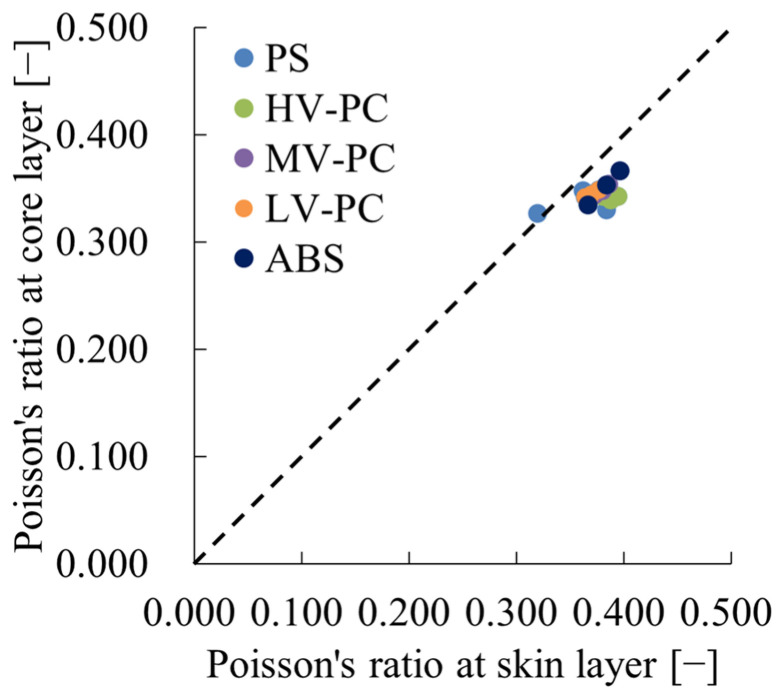
Relations between Poisson’s ratio at skin layer and that at core layer of amorphous polymers.

**Figure 8 polymers-16-01956-f008:**
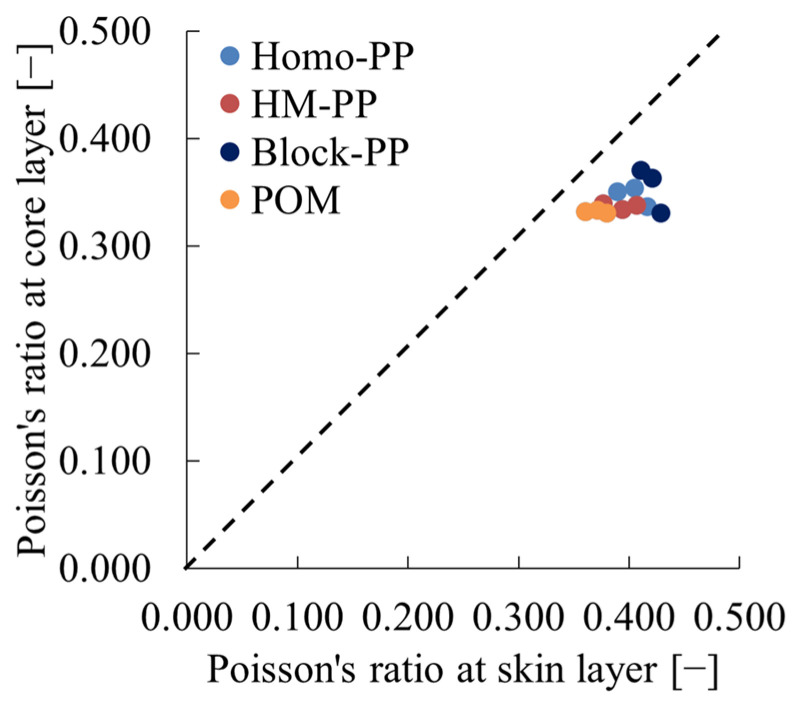
Relations between Poisson’s ratio at skin layer and that at core layer of crystalline polymers.

**Figure 9 polymers-16-01956-f009:**
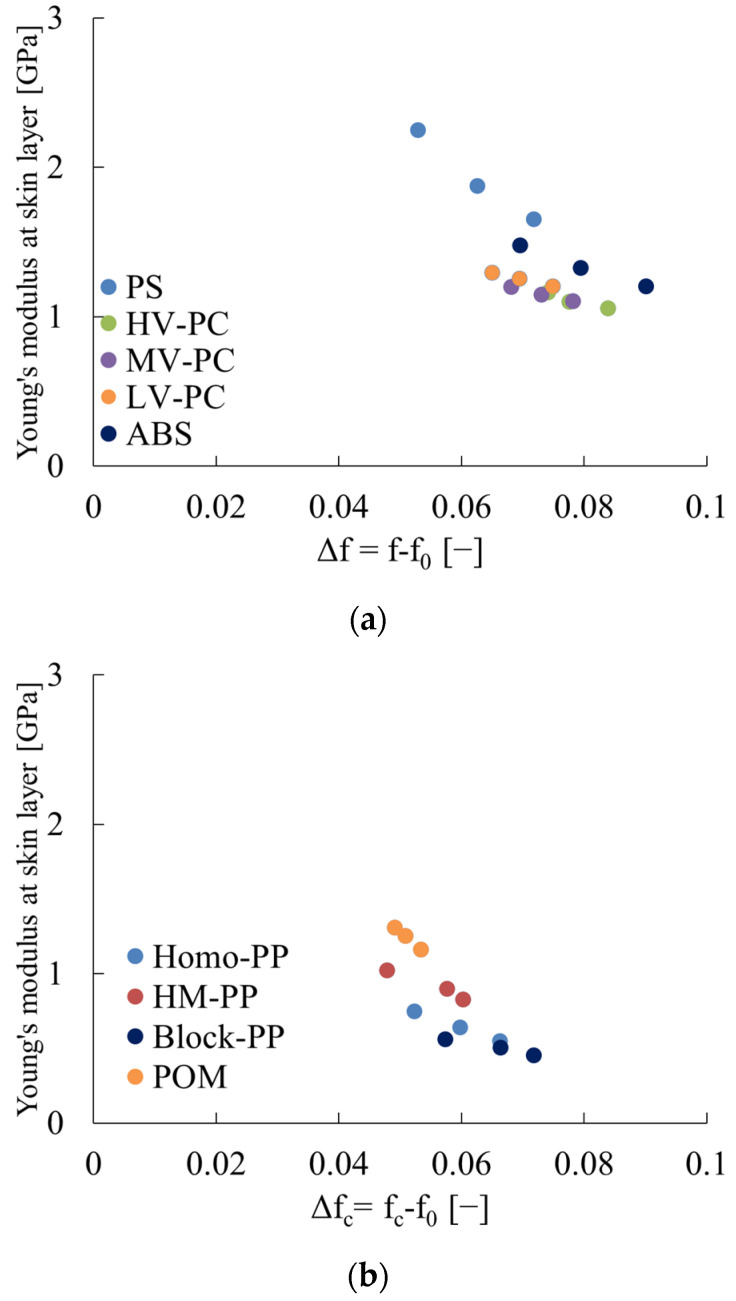
Relations between Young’s modulus and Δf or Δf_c_. (**a**) Amorphous polymer. (**b**) Crystalline polymer.

**Table 1 polymers-16-01956-t001:** Materials information. MVR in the table is melt volume flow rate.

Material	Code	Grade	Manufacture	MVR [cm^3^/10 min]
Polystyrene	PS	Toyo styrene GPPS G210C	Toyo Styrene Co., Ltd., Tokyo, Japan	9 @200 °C, 5 kgf
Low viscosity-Polycarbonate	LV-PC	Iupilon H-4000	Mitsubishi Engineering-Plastics Co., Tokyo, Japan	60 @300 °C, 1.2 kgf
Middle viscosity-Polycarbonate	MV-PC	Iupilon H-3000	Mitsubishi Engineering-Plastics Co., Tokyo, Japan	28 @300 °C, 1.2 kgf
High viscosity-Polycarbonate	HV-PC	Iupilon S-2000	Mitsubishi Engineering-Plastics Co., Tokyo, Japan	9 @300 °C, 1.2 kgf
Poly(acrylonitrile-butadiene-styrene)	ABS	Kralastic GA-101	Nippon A&L Inc., Osaka, Japan	26 @220 °C, 10.0 kgf
Homo-type Polypropylene	Homo-PP	NovatexPP MA1B	Japan Polypropylene Co., Ltd., Tokyo, Japan	20 @230 °C, 2.160 kgf
Block-type Polypropylene	Block-PP	NovatexPP BC03B	Japan Polypropylene Co., Ltd., Tokyo, Japan	30 @230 °C, 2.160 kgf
High modulus Polypropylene	HM-PP	NovatexPP MA3H	Japan Polypropylene Co., Ltd., Tokyo, Japan	10 @230 °C, 2.160 kgf
Polyoxymethylene	POM	Duracon NW-02LV	Polyplastics Co., Ltd., Tokyo, Japan	18 @190 °C, 2.160 kgf

**Table 2 polymers-16-01956-t002:** Injection molding conditions.

Code	T_inj_ [°C] Low/Middle/High	T_mold_ [°C]	V_inj_ [m/s]	P_hold_ [MPa]	t_inj_ [s]	t_cool_ [s]
PS	210/230/250	70	30	56	30	15
LV-PC	280/290/300	90	20	70	30	30
MV-PC	290/300/310	90	20	70	30	30
HV-PC	305/315/325	120	20	70	30	30
ABS	250/270/290	70	20	56	30	30
Homo-PP	210/230/250	50	30	56	30	15
Block-PP	210/230/250	50	30	56	30	15
HM-PP	210/230/250	50	30	56	30	15
POM	210/220/230	60	10	70	10	15

**Table 3 polymers-16-01956-t003:** Flexural properties and free volume of amorphous polymers.

Code	T_inj_ [°C]	F.S. [MPa]	F.M. [MPa]	υ_skin_ [−]	T_g_ [°C]	Δf [−]
PS	210	96.2	3198	0.320	99.7	0.053
	230	88.5	3161	0.362	99.6	0.063
	250	85.4	3156	0.384	100.3	0.072
LV-PC	280	78.8	2203	0.364	144.6	0.074
	290	79.1	2205	0.370	145.1	0.078
	300	78.8	2201	0.377	143.9	0.084
MV-PC	290	76.6	2142	0.373	148.0	0.068
	300	76.0	2136	0.380	147.7	0.073
	310	75.8	2131	0.386	147.1	0.078
HV-PC	305	77.8	2180	0.381	150.4	0.065
	315	76.4	2150	0.388	153.2	0.070
	325	75.5	2169	0.395	150.2	0.075
ABS	250	73.8	2556	0.367	105.0	0.070
	270	72.0	2533	0.384	104.4	0.080
	290	70.7	2510	0.397	102.3	0.090

**Table 4 polymers-16-01956-t004:** Flexural properties, crystallinities, and free volumes of crystalline polymers.

Code	T_inj_ [°C]	F.S. [MPa]	F.M. [MPa]	υ_skin_ [−]	T_m_ [°C]	Δf [−]	X_c_ [−]	Δf_c_ [−]
Homo-PP	210	40.0	1481	0.390	153.7	0.095	0.45	0.052
	230	38.7	1420	0.405	158.1	0.104	0.42	0.060
	250	37.4	1356	0.417	157.8	0.113	0.41	0.066
Block-PP	210	48.9	1868	0.377	165.9	0.091	0.48	0.048
	230	48.0	1833	0.394	166.2	0.101	0.43	0.058
	250	47.9	1862	0.407	167.3	0.110	0.45	0.060
HM-PP	210	32.0	1312	0.411	163.0	0.092	0.38	0.057
	230	31.8	1291	0.421	165.6	0.101	0.34	0.066
	250	31.7	1270	0.429	165.2	0.111	0.35	0.072
POM	210	56.5	2075	0.361	165.1	0.092	0.46	0.049
	220	56.6	2118	0.372	165.2	0.096	0.47	0.051
	230	55.9	2061	0.380	166.0	0.101	0.47	0.053

**Table 5 polymers-16-01956-t005:** τ_s_, τ_m_, and τ_l_ obtained from short-beam shear tests of amorphous polymers.

Code	T_inj_ [°C]	τ_s_ [MPa]	τ_m_ [MPa]	τ_l_ [MPa]	υ_core_ [−]	υ_skin_ [−]
PS	210	11.4	15.5	20.6	0.327	0.320
	230	9.5	12.3	15.7	0.348	0.362
	250	8.6	12.1	15.7	0.330	0.384
LV-PC	280	8.7	11.4	14.7	0.342	0.364
	290	8.8	11.5	15.0	0.339	0.370
	300	8.6	11.3	14.6	0.343	0.377
MV-PC	290	9.1	12.8	15.9	0.344	0.373
	300	9.4	12.1	15.4	0.348	0.380
	310	9.3	12.3	15.3	0.354	0.386
HV-PC	305	8.9	12.1	15.4	0.342	0.381
	315	8.9	12.7	15.7	0.344	0.388
	325	9.0	12.0	15.0	0.349	0.395
ABS	250	9.1	13.3	16.8	0.335	0.367
	270	9.2	12.7	15.5	0.353	0.384
	290	9.3	2.1	14.6	0.366	0.397

**Table 6 polymers-16-01956-t006:** τ_s_, τ_m_, and τ_l_ obtained from short-beam shear tests of crystalline polymers.

Code	T_inj_ [°C]	τ_s_ [MPa]	τ_m_ [MPa]	τ_l_ [MPa]	υ_core_ [−]	υ_skin_ [−]
Homo-PP	210	5.3	6.7	8.6	0.350	0.390
	230	5.3	6.6	8.4	0.354	0.405
	250	4.8	6.5	8.4	0.336	0.417
Block-PP	210	5.4	7.1	9.2	0.339	0.377
	230	5.3	7.1	9.3	0.334	0.394
	250	4.8	6.4	8.3	0.338	0.407
HM-PP	210	4.9	5.9	7.3	0.370	0.411
	230	4.8	5.8	7.3	0.363	0.421
	250	4.4	5.3	7.3	0.331	0.429
POM	210	7.0	9.2	12.2	0.332	0.361
	220	7.1	9.4	12.4	0.333	0.372
	230	6.7	8.9	11.8	0.331	0.380

**Table 7 polymers-16-01956-t007:** Tr obtained from Equation (20) of injection molded polymers.

Code	T_inj_ [°C] Low/Middle/High	υ_skin_ [−] Low/Middle/High	υ_core_ [−] Low/Middle/High	T_r_ [°C] Low/Middle/High
PS	210/230/250	0.320/0.362/0.384	0.327/0.348/0.330	6.3/7.6/43.1
LV-PC	280/290/300	0.364/0.370/0.377	0.342/0.339/0.343	36.8/46.7/56.9
MV-PC	290/300/310	0.373/0.380/0.386	0.344/0.348/0.354	22.9/29.4/34.4
HV-PC	305/315/325	0.381/0.388/0.395	0.342/0.344/0.349	18.2/25.4/32.6
ABS	250/270/290	0.367/0.384/0.397	0.335/0.353/0.366	34.0/37.7/46.9
Homo-PP	210/230/250	0.390/0.405/0.417	0.350/0.354/0.336	24.2/45.4/87.7
Block-PP	210/230/250	0.377/0.394/0.407	0.339/0.334/0.338	21.7/61.7/69.7
HM-PP	210/230/250	0.411/0.421/0.429	0.370/0.363/0.331	15.5/51.4/103.0
POM	210/220/230	0.361/0.372/0.380	0.332/0.333/0.331	34.6/40.7/53.5

## Data Availability

The original contributions presented in the study are included in the article, further inquiries can be directed to the corresponding authors.

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
