# Peer review of "Poisson’s Ratio Prediction of Injection Molded Thermoplastics Using Differential Scanning Calorimetry"

_polymers, 2024, doi:10.3390/polym16141956_

Round 1

Reviewer 1 Report

Comments and Suggestions for Authors

This manuscript establishes a link between Free volume and Poisson's ratio with novelty. The introduction is comprehensive and the experiments are detailed. It is recommended to be revised and published. The specific modifications are as follows:

1. For result 2.1. Free volume may be able to affect Poisson's ratio, however, why does the change in free volume, Δf establish a relationship with Poisson's ratio? Can you add more explanation here in terms of physical significance? Or in section 3.

2. For table 2. From table 2, it seems that increasing the moulding temperature can increase the melting point of PP, but why does it say "no clear correlation"?

3. For equation 7, please add a detailed description of the symbols in equation 7.

4. Why is the Experimental section in Part IV? Suggest to put it in the second part.

5. In the conclusion section, can you add the physical meaning of the correlation between Poisson's ratio and free volume, that is, the reason for the correlation?

Author Response

Paper No.: polymers-3087390

Cover Letter to Reviewer #1

Thank you for reviewing this paper. We will respond below to the items you have indicated.

  1. For result 2.1. Free volume may be able to affect Poisson's ratio, however, why does the change in free volume, Δf establish a relationship with Poisson's ratio? Can you add more explanation here in terms of physical significance? Or in section 3.

Thank you for pointing this out, I have added the reason for organizing by Δf.

  1. For table 2. From table 2, it seems that increasing the moulding temperature can increase the melting point of PP, but why does it say "no clear correlation"?

Thank you for pointing this out. It was a lack of interpretation on our part. We have corrected the description of the relevant results.

  1. For equation 7, please add a detailed description of the symbols in equation 7.

Thank you for pointing this out. I have added the necessary explanation to understand.

  1. Why is the Experimental section in Part IV? Suggest to put it in the second part.

Thank you for pointing this out. Since this paper was a transfer from another journal, the structure of the paper was different from Polymers. The structure has been corrected.

  1. In the conclusion section, can you add the physical meaning of the correlation between Poisson's ratio and free volume, that is, the reason for the correlation?

Thank you for pointing this out. I have added the physical meaning of this paper at the end.

Reviewer 2 Report

Comments and Suggestions for Authors

The paper by Tetsuo Takayama and Yuuki Nagasawa was the first to determine the relationships between the Poisson's ratio and the change in free volume when measured the glass transition temperature of crystalline and amorphous polymers. Also, the fact of Poisson's ratio decreases when approaching from the skin to the core layer was detected. This work is an important contribution to the field of physical and mechanical chemistry. The results of this work allow us to determine the mechanical characteristics of crystalline and amorphous polymers by methods of thermal analysis, which will significantly minimize the time of research and subsequent implementation of polymers.

However, the manuscript needs some minor revision in terms of writing. My specific comments are listed below.

1. Equation 6, there should be definitions of parameters τs, τm, and τl in the text.

2. The name of the DSC instrument, the type and kind of crucibles, the gas purge rate of the furnace, and the type of gas used should be given.

3. Section 4.3. It may be necessary to correct "10 mm/min" to "10 K/min".

4. In Figures 3, 4 and 5, the correlation coefficient should be included.

5. It is not quite clear how equation 7 was derived from equation 3 and 5. Perhaps equation 4 should be used?

6. In Table 7, the molecular weights of the polymers should be included.

I recommend to transfer the Materials and Methods section to the beginning of the article, right after the introduction.

Author Response

Paper No.: polymers-3087390

Cover Letter to Reviewer #2

Thank you for reviewing this paper. We will respond below to the items you have indicated.

  1. Equation 6, there should be definitions of parameters τs, τm, and τl in the text.
  • Thank you for pointing this out. We have changed the structure to make it easier to understand.

  1. The name of the DSC instrument, the type and kind of crucibles, the gas purge rate of the furnace, and the type of gas used should be given.
  • Thank you for pointing this out. We have added the items you pointed out to the best of our knowledge.

  1. Section 4.3. It may be necessary to correct "10 mm/min" to "10 K/min".
  • Thank you for pointing this out. We have corrected the units correctly.

  1. In Figures 3, 4 and 5, the correlation coefficient should be included.
  • Thank you for pointing this out. I have added the correlation coefficient.

  1. It is not quite clear how equation 7 was derived from equation 3 and 5. Perhaps equation 4 should be used?
  • Thank you for pointing this out. I have added a description of the formula.

  1. In Table 7, the molecular weights of the polymers should be included.
  • Thank you for pointing this out. We wanted to add this information as molecular weight, but the information was not disclosed. Therefore, we have added the information of Melt Volume Flow Rate (MVR), which is an index of melt viscosity that is said to depend on molecular weight.

  1. I recommend to transfer the Materials and Methods section to the beginning of the article, right after the introduction.
  • Thank you for pointing this out. Since this paper was a transfer from another journal, the structure of the paper was different from Polymers. The structure has been corrected.
